# Anti-Cancer Properties of Two Intravenously Administrable Curcumin Formulations as Evaluated in the 3D Patient-Derived Cancer Spheroid Model

**DOI:** 10.3390/ijms25158543

**Published:** 2024-08-05

**Authors:** Marlene Niederreiter, Julia Klein, Sebastian B. M. Schmitz, Jens Werner, Barbara Mayer

**Affiliations:** 1Department of General, Visceral and Transplant Surgery, Ludwig-Maximilians University Munich, Marchioninistraße 15, 81377 Munich, Germany; marlene.niederreiter@med.uni-muenchen.de (M.N.); julie-klein@gmx.net (J.K.); sebastian.schmitz@med.uni-muenchen.de (S.B.M.S.); jens.werner@med.uni-muenchen.de (J.W.); 2German Cancer Consortium (DKTK), Partner Site Munich, Pettenkoferstraße 8a, 80336 Munich, Germany; 3SpheroTec GmbH, Am Klopferspitz 19, 82152 Martinsried, Germany

**Keywords:** curcumin, curcumin formulations, 3D cancer spheroid model, solid tumors, patient derived, Kolliphor ELP, β-Cyclodextrin, polyethylene glycol

## Abstract

Curcumin (Cur) is a heavily used complementary derived drug from cancer patients. Spheroid samples derived from 82 patients were prepared and treated after 48 h with two Cur formulations (CurA, CurB) in mono- and combination therapy. After 72 h, cell viability and morphology were assessed. The Cur formulations had significant inhibitory effects of −8.47% (*p* < 0.001), CurA of −10.01% (−50.14–23.11%, *p* = 0.001) and CurB of −6.30% (−33.50–19.30%, *p* = 0.006), compared to their solvent controls Polyethylene-glycol, β-Cyclodextrin (CurA) and Kolliphor-ELP, Citrate (CurB). Cur formulations were more effective in prostate cancer (−19.54%) and less effective in gynecological non-breast cancers (0.30%). CurA showed better responses in samples of patients <40 (−13.81%) and >70 years of age (−17.74%). CurB had stronger effects in metastasized and heavily pretreated tumors. Combinations of Cur formulations and standard therapies were superior in 20/47 samples (42.55%) and inferior in 7/47 (14.89%). CurB stimulated chemo-doublets more strongly than monotherapies (−0.53% vs. −6.51%, *p* = 0.022) and more effectively than CurA (−6.51% vs. 3.33%, *p* = 0.005). Combinations of Cur formulations with Artesunate, Resveratrol and vitamin C were superior in 35/70 (50.00%) and inferior in 16/70 (22.86%) of samples. Cur formulations were significantly enhanced by combination with Artesunate (*p* = 0.020). Cur formulations showed a high variance in their anti-cancer effects, suggesting a need for individual testing before administration.

## 1. Introduction

Curcumin (Cur) is one of the most heavily used complementary substances by cancer patients [1]. Understanding the regulatory impact of Cur on carcinogenesis and tumor progression represents an intensely investigated subject in cancer research. A substantial number of analyses of cancer cell lines suggest that Cur has an impact on a variety of molecular signaling pathways [2,3,4,5]. In addition, induction of apoptosis was observed in patient-related tissues after the treatment of tumor patients with Cur [6,7]. These results promoted the analysis of Cur in several clinical trials. Cases of stable disease were observed in individual patients [8,9] and Cur led to chemo- and radiotherapy modulation [6,10,11] and the improvement of precancerous lesions [12,13,14]. In addition, Cur was found to ameliorate treatment-related side effects [15,16,17] and to improve quality of life in some patients [18,19]. However, these positive findings could not be confirmed by all clinical studies [20,21,22].

It is a commonly known fact that Cur is highly lipophilic, and therefore hardly bio-available after oral intake [23]. Therefore, numerous strategies have been developed to circumvent this issue. Examples are co-administration with adjuvants like Piperine or Lecithin, the incorporation into nanoparticles including micelles or liposomes, the usage of adjusted solvent components, and chemical structural changes and changes of route of administration, such as intravenous injection or mucosal application [24,25,26,27]. In consequence, a large number of Cur derivatives are available on the market. This raises the question of which Cur formulation benefits which patient. One possibility of gaining answers might be achieved through functional testing in 3D patient-related tumor models, e.g., organoid and spheroid cultures directly derived from patients’ cancer tissues. For example, after treatment with BCM-95 Cur (95% Curcuminoids) and amorphous Cur in colorectal cancer patient-derived organoids, significant decreases in organoid count, growth and viability were observed [28,29,30]. These data underline the urgent need for further investigations of different Cur formulations for the individual patient. Therefore, the objectives of this paper include the examination of Cur’s efficacy in relation to its formulation, patients’ clinical parameters, and its combination with other natural products. Furthermore, this paper focuses on the modulating impacts of Cur on standard therapies, as well as the effects of Cur’s solvent compounds.

This study tested two intravenously available Cur formulations and their correspondingly relevant solvents. Curcumin A (CurA) is a fully natural product, dissolved in β-Cyclodextrin and Polyethylene glycol (PEG) 400. Curcumin B (CurB), earlier introduced as CUC-1 [11], consists of 95% semi-synthetically produced Curcuminoids. The solvent for CurB is composed of Kolliphor ELP, ethanol and citric acid. Cur testing was performed using the patient-derived cancer spheroid model (PDCS), which was shown to be predictive for clinical drug response [31,32,33]. This model retains the key features of patients’ tumor tissues, including components of the tumor microenvironment [34]. Drug testing results are available within one week.

Cur was observed to be a drug with limited anti-cancer effects on PDCS, while its solvent compounds might additionally induce unwanted effects. Not all patient samples responded similarly to Cur treatment, but were dependent on distinct tumor characteristics. Combination with standard therapies resulted in both their stimulation and inhibition, while the efficacy of Cur was stimulated as well as inhibited by other natural substances. Therefore, these results seem to have a societal and industrial consequence.

## 2. Results

### 2.1. Patient Collective

The study collective included a total of 82 cancer patients (Table 1). Some patients contributed more than one sample or were tested comparatively with both Cur formulations. Therefore, a total of 106 samples were tested.

The majority (62.20%) of the patients were female and the median age was 58 years. The most common tumor entities in the study group were gynecological tumors (47.56%), followed by gastrointestinal cancers (18.29%) and rare tumors (15.85%). Most patients were either not pre-treated (39.02%) or heavily pre-treated (more than one pre-treated standard regimen, 39.02%). More than half of the patients (57.32%) had primary tumors, and were diagnosed in an advanced stage (65.85%).

### 2.2. Anti-Cancer Effects of Curcumin Formulations in PDCS

Curcumin formulations (Cur) were tested on 106 patient-derived cancer spheroid (PDCS) samples. They showed mean inhibitory effects of −8.47% ± 12.13%. PDCS cell viability was inhibited in 83/106 (78.30%) samples, while it was stimulated in 23/106 (21.70%) samples. 

Curcumin A (CurA) was tested on 62/106 (58.49%) patient samples, displaying a mean inhibitory effect of −10.01% ± 13.02% on PDCS. PDCS viability was inhibited in 49/62 (79.03%) samples (from −50.14% to −0.70%), but stimulated in 13/62 (20.97%) samples (from 0.10% to 23.11%). These effects were significantly different from its solvent control (SC CurA) (*p* = 0.001). SC CurA inhibited PDCS in 29/62 (46.77%) samples, while it had stimulatory effects in 33/62 (53.23%) samples. This led to a mean stimulation of 1.88% ± 17.37% (from −34.03% to 57.75%) (Figure 1A).

Curcumin B (CurB) was tested in 44/106 (41.51%) PDCS samples. It showed mean anti-cancer effects of −6.30% ± 10.51%. The inhibition of PDCS viability was observed in 34/44 (77.27%) samples (from −33.50% to −1.00%), while the stimulation of PDCS viability was observed in 10/44 (22.73%) samples (from 0.50% to 19.30%). SC CurB inhibited cell viability in 17/44 (38.64%) samples and stimulated it in 27/44 (61.36%) samples, leading to a stimulation of 6.28% ± 20.51% (from −28.71% to 75.58%), these effects being significantly different from those in CurB (*p* = 0.006, Figure 1A). 

The anti-cancer effects of CurA did not differ from those of CurB (*p* = 1.000). Similarly, no significant differences were observed between SC CurA and SC CurB (*p* = 1.000).

In 19 PDCS samples, Curcumin formulations A and B were tested comparatively. In 9/19 (47.37%) samples, CurA exhibited stronger inhibitory effects on PDCS than CurB (differences from 0.20% to 30.10%). Conversely, CurB had stronger inhibitory properties than CurA in 10/19 (52.63%) cases (differences from 0.30% to 25.88%) (Figure 1B). 

After therapy, PDCSs were checked microscopically for morphological changes. Conspicuously, in a few samples, both CurB- and SC CurB-treated spheroids showed alterations in cellular morphology in the periphery, appearing spike-like (Figure 2C,E). After treatment with a cell culture medium, either CurA or SC CurA, these specific morphological changes were not observed (Figure 2A,B,D).

### 2.3. Anti-Cancer Effects of Curcumin Formulations Depending on Patient Cohort Parameters

The effect of Cur was evaluated depending on patient characteristics, as described previously (Table 1). Overall, Cur was significantly more effective in prostate cancer-derived samples (mean −19.54%, n = 10, *p* = 0.038), which remained significant for CurA (*p* = 0.031, Figure 3E) but not for CurB. Furthermore, gynecological tumors including ovarian, cervical, vaginal and endometrial cancer responded significantly less notably to Cur formulations’ treatment (mean 0.30%, n = 10) (Figure 3E,F).

CurA revealed stronger anti-cancer effects in samples of patients < 40 years (mean −13.81%, n = 6, *p* = 0.226) and > 70 years (mean −17.74%, n = 8, *p* = 0.208) (Figure 3C). In contrast to CurA, the effects of CurB were stronger in heavily pre-treated patients (mean −9.61%, n = 20) compared to non-pre-treated patients (mean −3.06%, n = 19, *p* = 0.060) (Figure 3H). Additionally, CurB showed stronger anti-cancer activity in metastatic cancers (mean −8.62%, n = 27) compared to non-metastatic tumors (mean −2.60%, n = 17, *p* = 0.062, Table 2) (Figure 3J). There were no differences between samples derived from male or female patients (Figure 3A,B). Regarding CurA, there were no subgroup-differences dependent on the number of pre-treated standard regimens or the tumor status (Figure 3G,I). In contrast, CurB did not show a better or worse response depending on the patients’ age (Figure 3D).

### 2.4. Modulation of Standard Therapy through Curcumin Formulations

Cur formulations were combined with standard mono- and double therapies of various drug classes (Figure 4 and Figure 5). The combination of Cur and standard monotherapies was considered the best treatment option in 10/34 samples (29.41%) (green bars in Figure 6 and Figure 7). The best treatment option was defined as being superior to either agent on its own. However, in 6/34 samples (17.65%), the combination of Cur and standard monotherapies was considered the worst treatment option, being inferior to both agents on their own (red bars, Figure 6 and Figure 7). Regarding standard double therapies, the combination with Cur formulations resulted in the best treatment option in 10/13 (76.92%) samples and the worst in only 1/13 (7.69%) (Figure 6 and Figure 7).

Co-treatment with Cur formulations led to a more frequent stimulation of efficacy in standard double (10/13 samples, 76.92%) than standard monotherapies (15/34 samples, 44.12%) (Figure 4 and Figure 5). 

CurB enhanced the anti-cancer effects of standard double therapies significantly more strongly than monotherapies (mean −0.53% vs. mean −6.51%, *p* = 0.022). There was furthermore a significantly stronger enhancement of standard doublets by CurB (mean −6.51%) than CurA (mean 3.33%, *p* = 0.005) as CurB exclusively led to stimulatory effects on the standard treatment in 6/6 samples (Figure 7, Appendix A).

There was no preferential combination partner of Cur or its formulations identified.

### 2.5. Modulation of Curcumin Formulations by Other Complementary Compounds

Cur formulations were combined with either Artesunate (Art), Resveratrol (Res) or vitamin C (VitC) in 70 PDCS samples (Figure 8 and Figure 9). Their combination resulted in being superior to either agent alone in 35/70 (50.00%) samples (green bars, Figure 10 and Figure 11). However, they were considered the worst treatment option in 16/70 (22.86%) samples (red bars, Figure 10 and Figure 11).

The combination of CurA with Art or VitC was frequently more effective compared to each complementary substance alone (Figure 8 and Figure 10). Nevertheless, in a substantial number of samples, the combination of CurA with a second natural substance resulted in a decrease in CurA’s efficacy (22/50 samples, 44.00%). Regarding CurB, the combination with all complementary substances resulted more frequently in higher effects (13/20, 65.00%) than their monotherapies (CurB 3/20, 15.00%, other complementary substances 4/20, 20.00%). The impairment of the anti-cancer effect of CurB by combination therapies was rarer (5/20 samples, 25.00%).

Interestingly, Art increased Cur formulations and specifically CurB significantly (*p* = 0.020 and *p* = 0.031) by means of −7.81% and −8.04%, while neither Res nor VitC had similar effects on Cur or its formulations (Appendix A).

## 3. Discussion

Curcumin (Cur) is one of the most popular substances in complementary and alternative medicine, which is evident in its growing European market numbers [1]. Combined with its poor bioavailability, this has led to a variety of novel, modified formulations.

This study examined Cur’s anti-cancer effects and compared two intravenously available formulations, a fully natural, β-Cyclodextrin- and Polyethyleneglycol-based Cur (CurA) and a semi-natural Kolliphor-ELP-based Cur (CurB), in the 3D patient-derived cancer spheroid (PDCS) model.

Cur formulations were tested in PDCS samples from 82 patients. The anti-cancer effect was small but significant in comparison to the corresponding solvent controls. The comparably small anti-cancer efficacy of Cur as a single agent was also found in clinical trials [20]. In two interventional trials, Sharma et al. observed radiologically assessed stable disease in a few colorectal cancer patients, while no complete or partial response was detected [8,9]. 

Solvent controls of both Cur formulations had modulating impacts on spheroid viability. Despite consisting of different components, both solvent controls induced cell inhibition and stimulation. Similarly, in preclinical models, β-Cyclodextrin, Polyethyleneglycol (PEG), Kolliphor ELP or their families showed a variety of effects, i.e., induction of cell death in plenty of cancer cell lines [36,37,38]. In addition, benign immune cells were functionally modulated [39,40,41]. In the present study, changes in the cellular morphology were observed after treatment with CurB and its Kolliphor ELP-containing solvent control. After treatment with polyoxyethylated castor oil, drived from the cremophor family like Kolliphor ELP, changes in the shapes of blood cells have been described [42]. All substrates included in the tested solvent controls exert substantial side effects in patients. β-Cyclodextrin showed nephrotoxic impacts and hemolytic actions in former studies [37,43]. PEGs also exert renal toxicity at high doses, as well as allergic reactions [44]. Hypersensitivity reactions were furthermore observed for cremophores, including tachycardia, dyspnea and hypotension, mostly through the concentration-dependent activation of the complement system [45]. These profound changes induced by the described solvents underline the need for the testing of corresponding solvent controls in the experimental and clinical study setting in separate control groups.

This study used two intravenously administrable Cur formulations, as intravenous application seems to yield higher peak plasma concentrations compared to oral intake [46,47]. The latest developments also include mucosal uptake of Cur through sprays [27] or fused tablets. The further enhancement of Cur’s efficacy was attempted by many strategies, including combination with other complementary compounds. Its combination with pepper fruit and ginger [48], Omega-3 fatty acids [49], and Soy Lecithin [50] was investigated in clinical studies, to name only a few.

In this trial, Cur was combined with Artesunate (Art), Resveratrol (Res) and vitamin C (VitC), which pre-clinically exert anti-cancer effects themselves [51,52,53]. The combination of Cur and another complementary substance was considered the best treatment option in 50% of samples tested, being superior to both agents alone. Significant improvement, however, was only observed in CurB combined with Art. Conversely, in 22.86% of samples, the combination of Cur and its complementary partners was considered the worst treatment option, being inferior to each individual therapy alone.

This study also combined Cur with standard mono- and double therapies. This combination was considered superior in comparison to either agent alone in almost 45%, but inferior in approximately 15% of samples tested. All mean enhancement effects remained under 10%. Relatable results were found in clinical studies that tested Cur in combination with standard treatment regimens, including anti-cancer drugs and radiotherapy. Significant improvements in Objective Response Rates (ORR) were observed in some randomized–controlled trials (RCTs) on solid cancers [10,11,20]. The Cur formulation that this study called CurB was tested intravenously by Saghatelyan et al. [11] in a randomized–controlled trial. Breast cancer patients received CurB in combination with Paclitaxel for 12 weeks, reaching significantly higher ORR in comparison to patients receiving Paclitaxel and placebo. Furthermore, a clinical trial with cervical cancer patients observed significantly more cases of complete tumor response after Cur intake for several days in combination with radiation therapy [6], suggesting radio-sensitizing effects.

Regarding overall (OS) and progression-free survival (PFS), however, observations of RCTs were mostly discouraging. The significant improvement of OS was observed in only one RCT [10,20], and of PFS in none. Based on this lack of evidence, the statements of the American Oncology Nursing Society [54] and the German Association of the Scientific Medical Societies [55] could not give any recommendations in their guidelines for or against the use of Cur in standard cancer treatment. 

As the number of Cur formulations and derivatives grows each year, so too does the necessity to evaluate which patient might benefit from which type of Cur. This study examined connections between the efficacy of Cur formulations and biological gender, age, tumor entity, number of pre-treated standard regimens and tumor status. While there were no differences regarding biological gender, CurA seemed to be slightly more effective in samples of patients over 70 years of age. Elder patients in particular are often confronted with a reduction in chemotherapy, either upfront or during the treatment course, due to drug-induced side effects [56,57]. In this situation, CurA could be used as an add-on therapy. This is further supported by chemo-stimulatory effects induced by Cur that were observed in individual patients in this study.

This trial observed that some cancer types are more susceptible towards Cur than others. CurA was shown to be more effective in prostate cancer-derived spheroids. Choi et al. [58] showed a significantly lower proportion of patients with prostate-specific antigen (PSA) progression during the oral intake of Cur. However, changes of PSA levels themselves were not significant, which other clinical trials also observed [59,60]. Interestingly, there was also a trend in gynecological tumors (breast cancers excluded) towards a worse response after CurA or CurB treatment. Instead of using the here-tested Cur formulations, there are indications that monotherapies with Art and VitC might be alternative treatment options from the field of natural substances in premalignant lesions and malignant gynecological tumors [61,62].

Regarding the number of pre-treated standard regimens, CurA showed similar outcomes in all cohort groups, while CurB showed a trend towards an increasing effect with an increasing number of pre-treated regimens. This leads to the assumption that one would use CurA mostly in non- or slightly pre-treated patients, while CurB seems to exert its most favorable effects in heavily pre-treated ones. In terms of tumor status, CurA showed steady inhibitory effects, especially in the primary tumor situation, while CurB showed increasing effects with the further advancement of tumor burden in cases of both primary and recurrent tumors. Considering this, CurA is probably more favorable in the primary and recurrent locally restricted and advanced situations, while CurB should rather be considered in the metastasized situation of recurrent tumors. In consequence, CurA treatment seems appropriate for other patient subgroups compared to CurB, and vice versa.

Understanding the mechanisms that lead to stronger Cur responses in some cancers than in others would be helpful for patient stratification and the development of new treatment options. However, almost all information concerning Cur-involved pathways was gained in cancer cell lines [2,3,4,5], not reflecting the complexity of patients’ tissues. Therefore, these findings are hardly translatable into clinical use. In addition, the microenvironmental components, such as fibroblasts and immune cells, might have further modulatory impacts. Due to this heterogeneity, it seems worthwhile to characterize individual cancers responding to a certain Cur formulation in depth to identify responsible mechanisms.

This study tested two Cur formulations and their corresponding solvent controls under the same conditions in a 3D patient-derived cancer spheroid model. The limitations of this study include the rather small number of patients, especially regarding subgroup analyses, although the patient collective of more than 80 patients is larger than those in many clinical trials investigating Cur. As the half-life of Cur in aqueous solutions is short, it might also be necessary to reapply the substance during the treatment period. In addition, this paper observed a strong variance in the anti-cancer effects of Cur formulations, which might be connected to inter-individual differences in patients’ tumors.

## 4. Materials and Methods

### 4.1. Patient Characteristics and Cancer Tissue

For each patient, biological gender, age, tumor entity, number of pre-treated standard regimens and tumor status were documented. Rare cancers were defined by an incidence lower than 6/100,000 [35]. Tumor status was distinguished into primary and recurrent tumors, either locally restricted, locally advanced or metastasized (Table 3).

Tissues used in this study were provided by the Biobank of the Department of General, Visceral and Transplantation Surgery at Ludwig Maximilians University (LMU) under the administration of the Human Tissue and Cell Research (HTCR) Foundation. The framework of the HTCR Foundation, which includes obtaining written informed consent from all donors, has been approved by the ethics commission of the Faculty of Medicine at the LMU (approval number 025-12, 14 January 2014) in Germany. From every cancer patient, at least one tissue sample was tested. Due to multiple malignant localizations, there were three patients from whom two tissue samples were used and one more patient from whom three samples were available.

### 4.2. Spheroid Formation

Patient-derived cancer spheroids (PDCS) were prepared directly from fresh cancer tissues, as described earlier [51]. Briefly, tumor tissues were suspended through mechanical and enzymatic digestion, using a Liberase enzyme mixture (Roche, Mannheim, Germany) in accordance with the manufacturer’s protocol. Cell number and viability were determined using the Trypan blue exclusion test. Suspended cells were seeded at a density of 50,000 cells per well in a low-adhesion round-bottom 96-well plate, and left for 48 h for spheroid formation under standard cell culture conditions.

### 4.3. Curcumin Formulations and Treatment

After formation, PDCSs were treated with two different Curcumin (Cur) formulations for 72 h under standard cell culture conditions (37 °C, 5% CO_2_). There were no washing steps or medium changes performed to the imposition of avoid mechanical stress on the spheroids by disturbance. Both Curcumins had a purity of 98–100%. Curcumin A (CurA) was obtained from Arnika Apotheke, Unterhaching, Germany. Curcumin B (CurB), earlier introduced as CUC-1 [11] was obtained from Burg Apotheke, Königstein im Taunus, Germany. The abbreviation Cur includes both formulations CurA and CurB. Both Cur formulations and their corresponding solvents were tested in the clinically achievable peak plasma concentration of 2.36 µg/mL after intravenous injection [46,47]. In addition, the modulating effects of the Cur formulations on conventional anti-cancer drugs were evaluated in clinically achievable concentrations. The polyphenols were combined with chemotherapeutics, hormonal therapies, immune therapies, and targeted therapies using mono- and double therapies, depending on the practitioners’ recommendations for the individual patient. Moreover, Cur formulations were combined with other frequently applied natural substances, namely, Artesunate (0.74 µg/mL), Resveratrol (0.44 µg/mL) and vitamin C (3.43 µg/mL). All of them were obtained from Burg Apotheke. In most samples, therapeutics were tested comparatively. For combination therapies with anti-cancer drugs and natural compounds, combination therapy has been considered the best treatment option when superior to both combination partners tested on their own. The worst treatment option was defined as being inferior, accordingly. Wanted and unwanted effects of the Cur formulations on cell viability and the morphology of the spheroids were assessed. The effects of Cur formulations and their corresponding solvent controls on cell viability were quantitively assessed using the CellTiter Glo Viability Assay (Promega, Walldorf, Germany), performed in accordance with the manufacturer’s protocol and read out using the FilterMax F3 luminometer (Molecular Devices, San Jose, CA, USA). Morphological changes induced by Cur formulations and their corresponding solvent controls were assessed using light-field microscopy.

### 4.4. Statistical Analysis

The anti-cancer effects of CurA and CurB were tested for normal distribution using Shapiro–Wilk’s test. After not fulfilling the parametric conditions, the effects were tested for significance with the Mann–Whitney U test or non-parametric ANOVA (Kruskal–Wallis test) and corrected for multiple testing using Bonferroni’s correction. Correlations between clinical parameters and Cur efficacy were analyzed using the 2-tailed Fisher’s exact test. Cut-off values were the means of Cur overall, CurA and CurB. Significant results were expected at *p*-values < 0.05, and marginally significant results at *p*-values between 0.05 and 0.1. The results have been shown as means ± standard deviation, and the range is given in parenthesis. All statistical calculations were carried out using SPSS Statistics (IBM, Armonk, NY, USA, Version 29). All figures displayed in this paper were designed with GraphPad Prism (Version 9.1.2).

## 5. Conclusions

This paper examined the anti-cancer effects of two Curcumin (Cur) formulations in the patient-derived cancer spheroid model (PDCS). It was found that both formulations had inhibitory as well as stimulatory effects on PDCS samples as well as their corresponding solvent controls. Depending on patients’ tumor characteristics, Cur formulations revealed differences in terms of their anti-cancer activities. Cur led to an increase as well as a decrease in the efficacy of standard therapeutics, and was individually modulated by other natural substances. Taken together, the present data illustrate the need for a pre-therapeutic testing of Cur formulations for each individual patient to ensure the benefits of Cur-based therapy. 

## Figures and Tables

**Figure 1 ijms-25-08543-f001:**
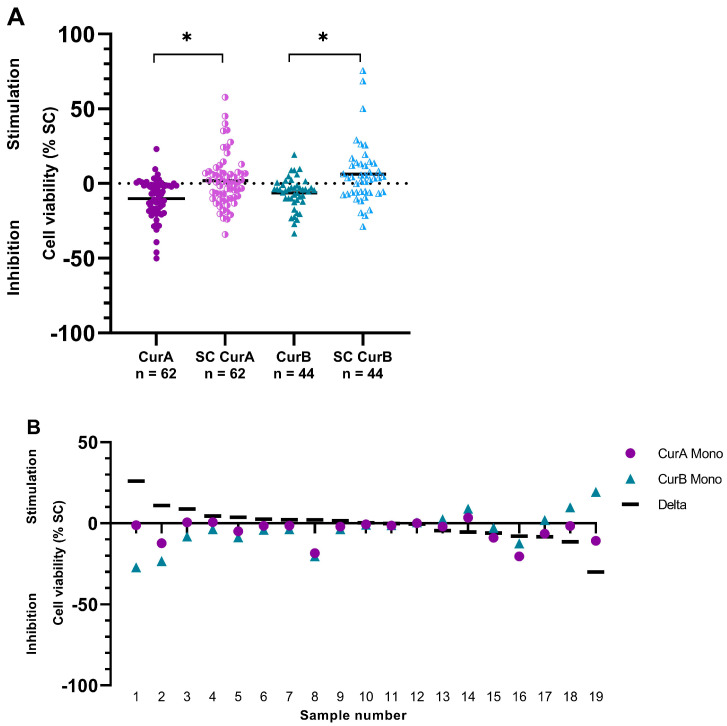
(**A**) Effects of Curcumin formulations A and B and their Solvent Controls (SC) in the patient-derived cancer spheroid model (PDCS). (**B**) Comparative testing of Curcumin formulations A and B in the same PDCS samples. Delta shows the difference between the two formulations. CurA, Curcumin A; CurB, Curcumin B; SC CurA, Solvent Control CurA; SC CurB, Solvent Control CurB; CurA Mono, monotherapy of Curcumin A; CurB Mono, monotherapy of Curcumin B. * *p* < 0.05.

**Figure 2 ijms-25-08543-f002:**
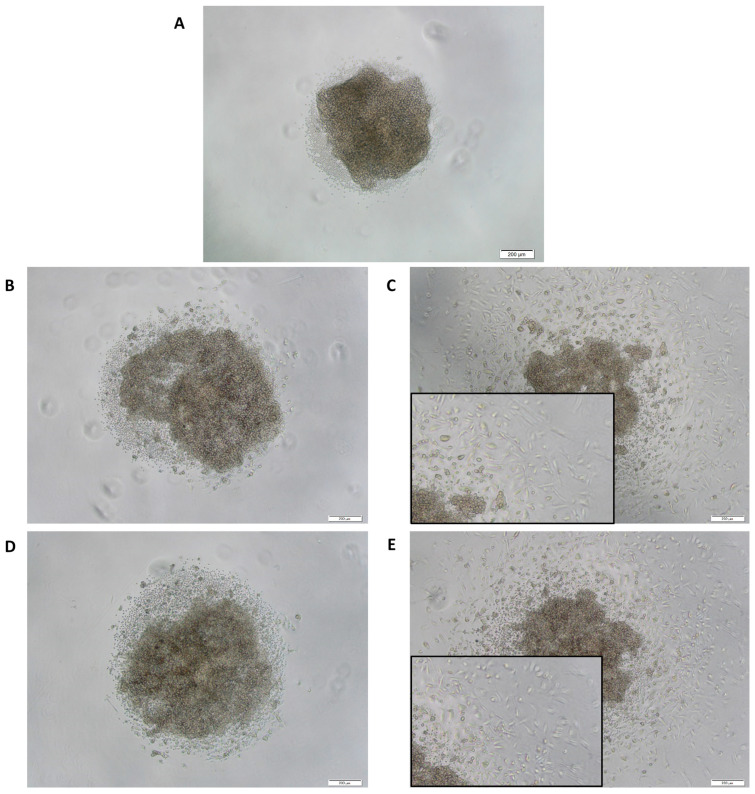
Cell morphology of patient-derived cancer spheroids (PDCS) of the same tumor sample after treatment with cell culture medium (**A**), Curcumin formulations A (**B**) and B (**C**), and their solvent controls (SC) ((**D**), SC CurA; (**E**), SC CurB). Large pictures are displayed at 40× magnification; smaller pictures show exemplary cells at 100× magnification. Scale bars in the bottom right corners display 200 µm.

**Figure 3 ijms-25-08543-f003:**
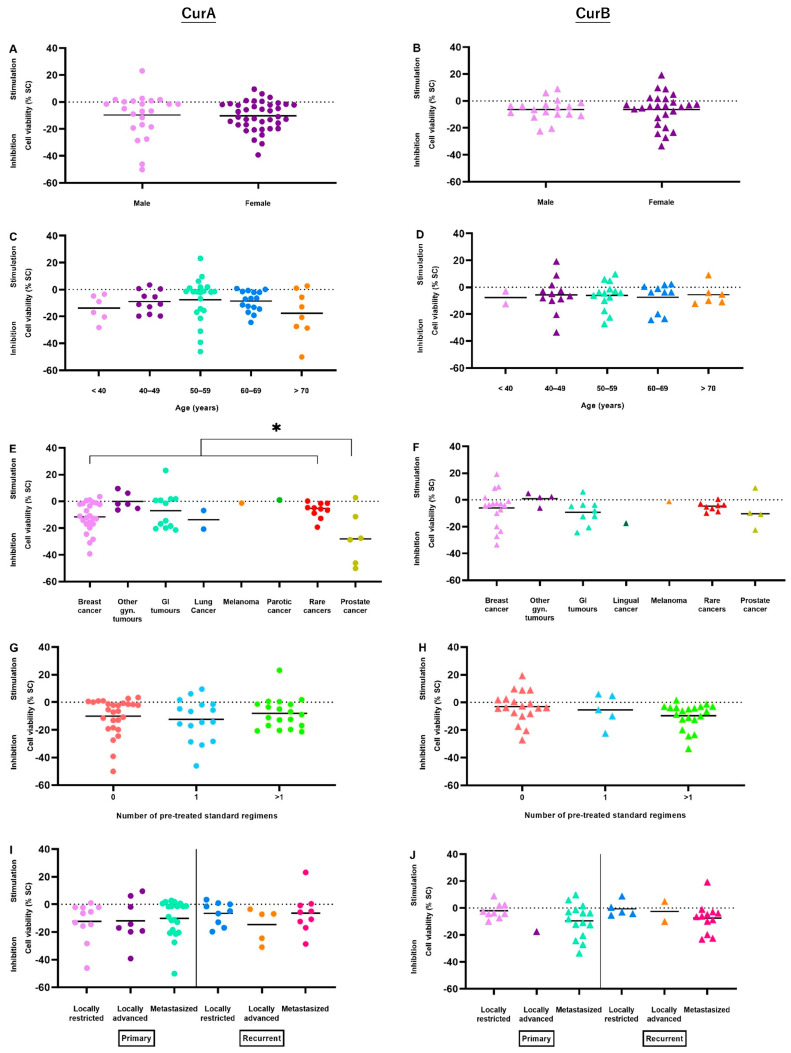
Anti-cancer effects of Curcumin formulations Curcumin A (**A**,**C**,**E**,**G**,**I**) and Curcumin B (**B**,**D**,**F**,**H**,**J**) depending on patient collective parameters biological gender (**A**,**B**), age (**C**,**D**), tumor entity (**E**,**F**), number of pre-treated standard regimens (**G**,**H**) and tumor status (**I**,**J**). CurA, Curcumin A; CurB, Curcumin B; Gyn, gynecological; GI, gastrointestinal. * *p* < 0.05.

**Figure 4 ijms-25-08543-f004:**
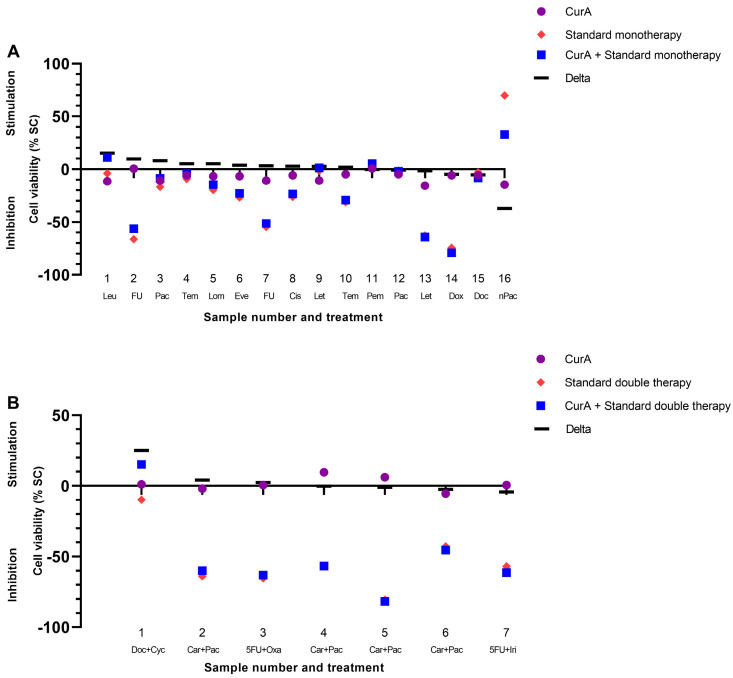
Combination of Curcumin A with standard mono- (**A**) and double therapies (**B**) in the patient-derived cancer spheroid (PDCS) model. Delta shows the difference between the standard treatment and its combination with CurA. (**A**) CurA, Curcumin A; Leu, Leuprorelin; FU, 5-Fluorouracil; Pac, Paclitaxel; Tem, Temozolomide; Lom, Lomustine; Eve, Everolimus; Cis, Cisplatin; Let, Letrozol; Pem, Pembrolizumab; Dox, Doxorubicin; Doc, Docetaxel; nPac, Nab-Paclitaxel. (**B**) Doc+Cyc, Docetaxel+Cyclophosphamid; Car+Pac, Carboplatin+Paclitaxel; 5FU+Oxa, 5-Fluorouracil+Oxaliplatin; 5FU+Iri, 5-Fluorouracil+Irinotecan.

**Figure 5 ijms-25-08543-f005:**
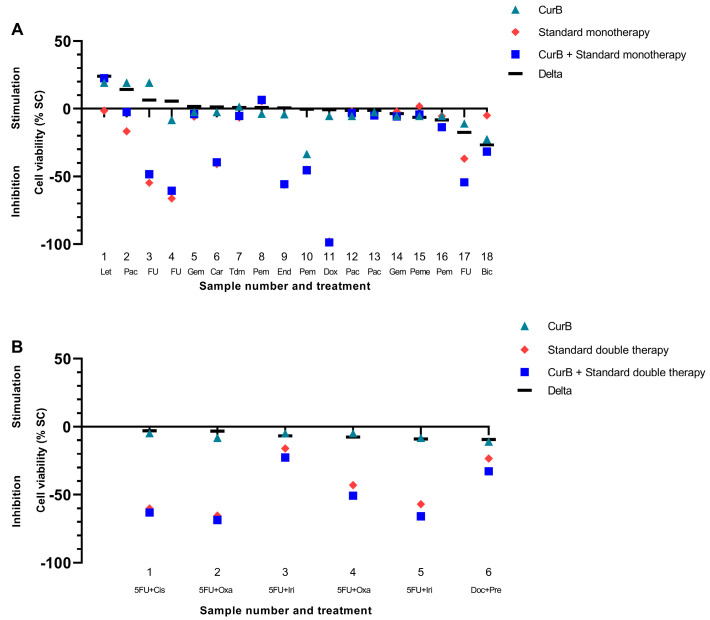
Combination of Curcumin B with standard mono- (**A**) and double therapies (**B**) in the patient-derived cancer spheroid (PDCS) model. Delta shows the difference between standard treatment and its combination with CurB. (**A**) CurB, Curcumin B; Let, Letrozol; Pac, Paclitaxel; FU, 5-Fluorouracil; Gem, Gemcitabine; Car, Carboplatin; Tdm, TDM-1; Pem, Pembrolizumab; End, Endoxifen; Dox, Doxorubicin; Pac, Paclitaxel; Peme, Pemetrexed, Bic, Bicalutamide. (**B**) 5FU+Cis, 5-Fluorouracil+Cisplatin; 5FU+Oxa, 5-Fluorouracil+Oxaliplatin; 5FU+Iri, 5-Fluorouracil+Irinotecan; Doc+Pre, Docetaxel+Prednisolon.

**Figure 6 ijms-25-08543-f006:**
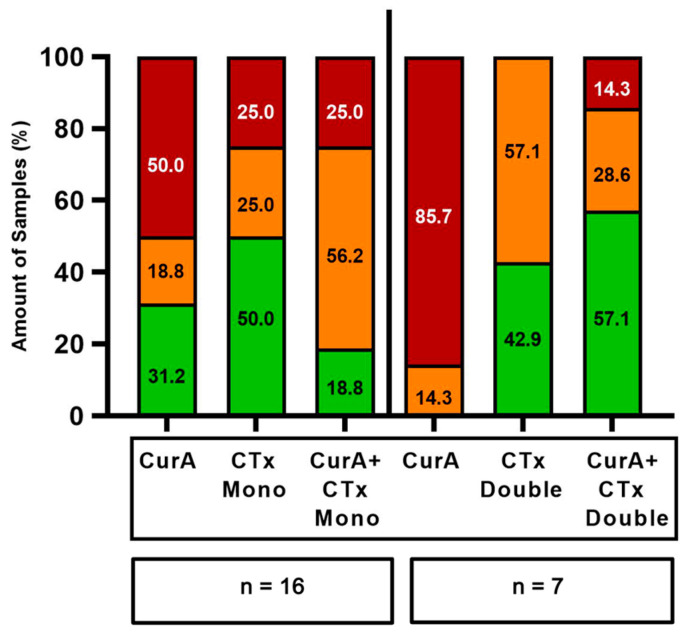
Frequency of the best and worst treatment options of Curcumin A, standard mono- and double therapies alone and their combinations. Red bars, worst treatment option; orange bars, neither worst nor best treatment option; green bars, best treatment option. CurA, Curcumin A; CTx Mono, standard monotherapies; CTx Double, standard double therapies; n, number of samples tested.

**Figure 7 ijms-25-08543-f007:**
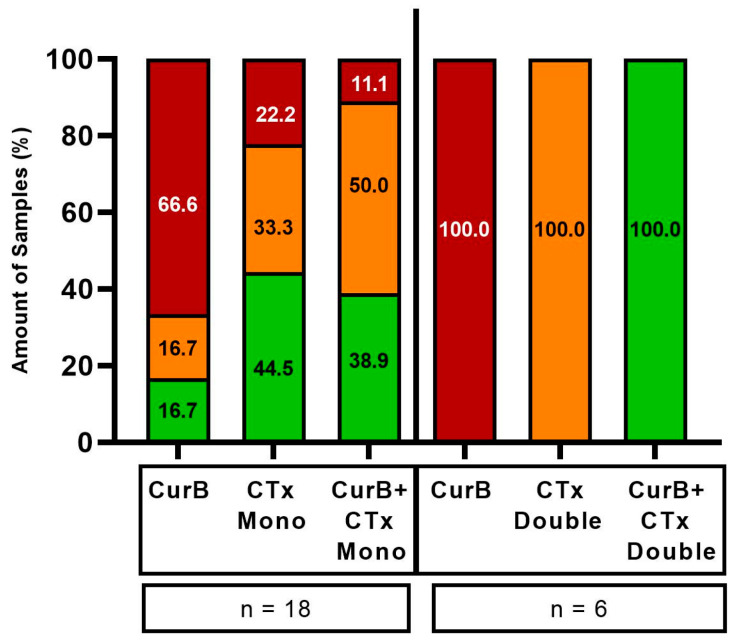
Frequency of the best and worst treatment options of Curcumin B, standard mono- and double therapies alone and their combination. Red bars, worst treatment option; orange bars, neither worst nor best treatment option; green bars, best treatment option. CurB, Curcumin B; CTx Mono, standard monotherapies; CTx Double, standard double therapies; n, number of samples tested.

**Figure 8 ijms-25-08543-f008:**
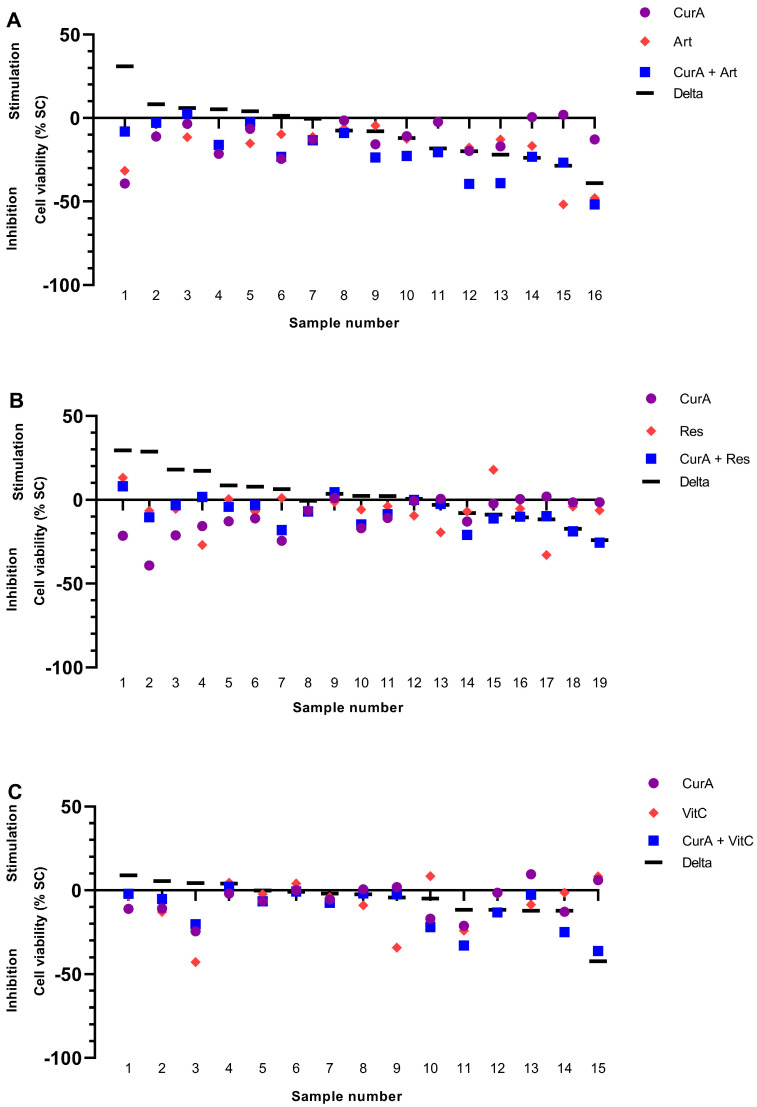
Combination of Curcumin A with Artesunate (**A**), Resveratrol (**B**) and Vitamin C (**C**) in the patient-derived cancer spheroid (PDCS) model. Delta shows the difference between CurA alone and in combination with a second complementary substance. CurA, Curcumin A; Art, Artesunate; Res, Resveratrol; VitC, vitamin C.

**Figure 9 ijms-25-08543-f009:**
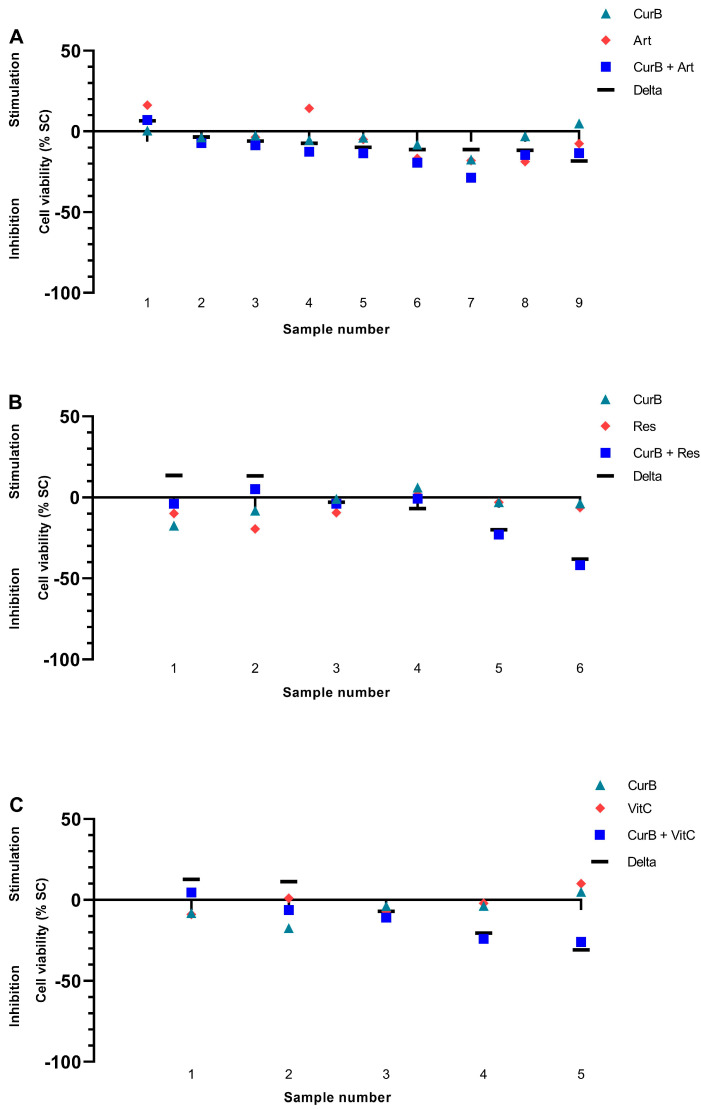
Combination of Curcumin B with Artesunate (**A**), Resveratrol (**B**) and Vitamin C (**C**) in the patient-derived cancer spheroid (PDCS) model. Delta shows the difference between CurB alone and in combination with a second complementary substance. CurB, Curcumin B; Art, Artesunate; Res, Resveratrol; VitC, Vitamin C.

**Figure 10 ijms-25-08543-f010:**
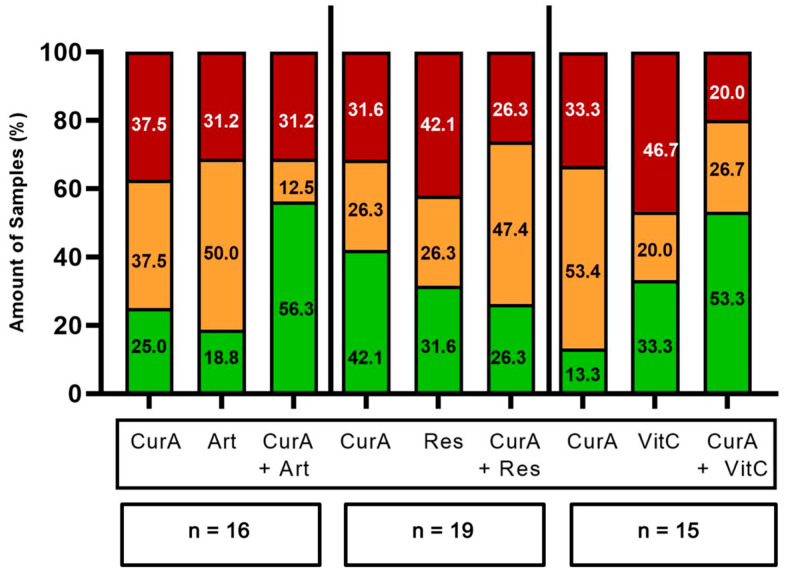
Frequency of the best and worst treatment options of Curcumin A and Art, Res and VitC alone and their combination. Red bars, worst treatment option; orange bars, neither worst nor best treatment option; green bars, best treatment option. CurA, Curcumin A; Art, Artesunate; Res, Resveratrol; VitC, Vitamin C; n, number of samples tested.

**Figure 11 ijms-25-08543-f011:**
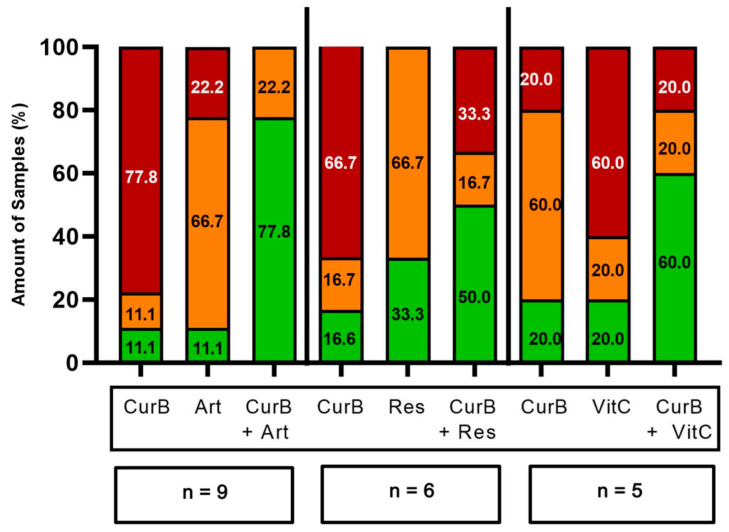
Frequency of the best and worst treatment options of Curcumin B and Art, Res and VitC alone and their combination. Red bars, worst treatment option; Orange bars, neither worst nor best treatment option; Green bars, best treatment option. CurB, Curcumin B; Art, Artesunate; Res, Resveratrol; VitC, Vitamin C; n, number of samples tested.

**Table 1 ijms-25-08543-t001:** Patient collective.

Parameter		N = 82
**Biological Gender**	Male	31
	Female	51
**Age (years)**		
	<40	6
	40–49	17
	50–59	27
	60–69	18
	>70	14
**Tumor entity**	Gynecological Tumors	39
	Breast Cancer	33
	Ovarian Cancer	3
	Cervical Cancer	1
	Vaginal Cancer	1
	Endometrial Cancer	1
	Gastrointestinal Tumors	15
	Colorectal Cancer	8
	OGJ adenocarcinoma	2
	Esophageal Cancer	1
	Gastric Cancer	1
	Appendix Neoplasia	1
	Papilla Vateri Cancer	1
	Pancreatic Cancer	1
	Rare Cancers ^#^	13
	Sarcoma	6
	Glioblastoma	2
	Peritoneal Mesothelioma	1
	Hepatoblastoma	1
	Pleural Fibroma	1
	Pleural Mesothelioma	1
	Bladder Cancer	1
	Others	15
	Prostate Cancer	10
	Lung Cancer	2
	Melanoma	1
	Parotic Cancer	1
	Lingual Cancer	1
**Number of pre-treated standard regimens ***	0	32
	1	18
	>1	32
**Tumor status**	Primary tumors	47
	Locally restricted	17
	Locally advanced	7
	Metastasized	23
	Recurrent tumors	35
	Locally restricted	11
	Locally advanced	7
	Metastasized	17

* Only standard drug therapy was considered; ^#^ as defined by [35]; OGJ, esophageal-gastric junctional; N, number of patients.

**Table 2 ijms-25-08543-t002:** Subgroup analyses dependent on patient characteristics.

			Subgroup Analyses									Fisher’s Exact Test *p*-Value		
			Cur Overall			CurA			CurB			Cur Overall	CurA	CurB
			Mean (%)	n	*p*-Value	Mean (%)	n	*p*-Value	Mean (%)	n	*p*-Value			
**Biological gender**	Male		−8.16	41	0.58	−9.60	23	0.36	−6.33	18	0.63	0.692	0.29	0.36
	Female		−8.67	65		−10.26	39		−6.27	26				
	Median	58												
**Age (years)**	≤58		−7.87	64	0.58	−9.15	37	0.46	−6.12	27	1.00	0.843	0.80	0.75
	>58		−9.38	42		−11.29	25		−6.58	17				
		<40	−12.27	8		−13.81	6		−7.67	2				
		40–49	−7.23	23		−8.96	11		−5.64	12				
		50–59	−7.01	35		−7.60	21		−6.14	14				
		60–69	−8.18	26		−8.63	16		−7.47	10				
		>70	−12.52	14		−17.74	8		−5.56	6				
**Tumor entity**	Breast cancer		−9.35	41	0.51	−11.68	24	0.32	−6.06	17	0.58	0.544	0.12	0.75
	Non-Breast cancer		−7.91	65		−8.96	38		−6.44	27				
		Breast cancer	−9.35	41		−11.68	24		−6.06	17				
		Other gynecological tumors *	0.30	10		−0.03	6		0.79	4				
		Gastrointestinal tumors	−8.03	21		−7.07	12		−9.31	9				
		Prostate cancer	−19.54	10		−26.86	6		−8.56	4				
		Lung cancer	−13.79	2		−13.79	2		x	x				
		Melanoma	−1.30	2		−1.40	1		−1.20	1				
		Parotic cancer	0.92	1		0.92	1		x	x				
		Lingual cancer	−17.47	1		x	x		−17.47	1				
		Rare cancers	−5.93	18		−6.62	10		−5.06	8				
**Number of pre-treated standard regimens**	0		−7.11	46	0.11	−9.96	27	0.57	−3.06	19	0.06	0.166	0.61	0.36
	≥1		−9.51	60		−10.06	35		−8.76	25				
		0	−7.11	46		−9.96	27		−3.06	19				
		1	−10.72	21		−12.41	16		−5.33	5				
		> 1	−8.86	39		−8.08	19		−9.61	20				
**Tumor status**	Non-metasta-sized		−8.12	50	0.65	−10.96	33	0.42	−2.60	17	0.06	0.326	1.00	0.11
	Metasta-sized		−8.79	56		−8.93	29		−8.62	27				
		Primary locally restricted	−7.70	20		−12.29	11		−2.08	9				
		Primary locally advanced	−12.57	9		−11.96	8		−17.47	1				
		Primary metastasized	−9.87	35		−10.12	20		−9.53	15				
		Recurrent locally restricted	−4.35	14		−6.44	9		−0.59	5				
		Recurrent locally advanced	−11.14	7		−14.58	5		−2.53	2				
		Recurrent metastasized	−6.97	21		−6.29	9		−7.49	12				

* Other gynecological tumors included ovarian cancer, cervical cancer, vaginal cancer and endometrial cancer; Cur, Curcumin; CurA, Curcumin A; CurB, Curcumin B; n, sample number in corresponding subgroup; x, not tested in corresponding subgroup.

**Table 3 ijms-25-08543-t003:** Definition of tumor status.

Tumor Status	Tumor Infiltration into Neighbor Organs	Local Lymph Node Metastasis	Distant Metastasis
Locally restricted	--	--	--
Locally advanced	X	--	--
	--	X	--
	X	X	--
Metastasized	--	--	X
	X	--	X
	--	X	X
	X	X	X

X, present; --, absent.

## Data Availability

The data presented in this study are available on request from the corresponding author.

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
