# Peer review of "Anti-Cancer Properties of Two Intravenously Administrable Curcumin Formulations as Evaluated in the 3D Patient-Derived Cancer Spheroid Model"

_ijms, 2024, doi:10.3390/ijms25158543_

Round 1
Reviewer 1 Report
Comments and Suggestions for Authors
The submitted article "Anti-cancer properties of two intravenously administrable Curcumin formulations as evaluated in the 3D patient-derived cancer spheroid model" is partially well presented and organized with original considerations. The issue is highly pertinent for the scientific community and society at large. The article should certainly be published after some minor revisions. However, I have a few suggestions and concerns that should be taken into account:
-Please provide a well-structured and comprehensive abstract and specify the methodology used to compile this research article.
-The introduction should be moderately improved. It must represent the outline of your manuscript by specifying the objectives and the necessary data. Please also mention the importance of this study for society and industry.
-The choice of these curcumin-formulations and combined therapies needs to be justified.
-The protocol of Cur-formulations tested in PDCS must be clearly described (e.g., the duration of treatment/under what circumstances).
- What was your assessment of the possible side effects of these formulations (Cur A and Cur B) for the different types of cancer tested?
- Table S1 should be included in the article's main content, as it summarizes all the results obtained.
-What are the suggested molecular pathways that could be targeted by these curative formulations, responsible for better responses in certain types of cancer in the patients tested? This question should be addressed in the discussion section.
- I strongly recommend underlining the limitations of this investigation.
-The conclusion section needs to be improved
- Abbreviations should be revised. They should be indicated in the first indication of the text, including in the abstract. Giving a list of abbreviations after the conclusion would be better.
-Ethics approval number must be provided.
Comments on the Quality of English LanguageMinor editing of English language required
Reviewer 2 Report
Comments and Suggestions for Authors
The paper investigates effects of curcumin formulations on patient-derived cancer spheroid model. The topic is very important, and the idea is excellent, however, there are some doubts about the study. As the same diagnosis does not mean that the outcome is similar, combining tumors of different origins in one group could result in loss of significance, especially as the authors state that curcumin is not so effective in gynecological tumors.
The paper holds a great idea, but methodologically it is not sufficient. The paper is more appropriate for clinical journal rather than IJMS.
In the abstract authors state that curcumin is “less effective in gynecological tumors, breast cancers excluded”, but they present as “gynecological tumors” 33 breast cancer, 3 ovarian, 1 of cervical, vaginal and endometrial tumors. 6 different gynecological tumors are not enough to perform the statistics.
Why did the authors included tumors which they had only one patient?
Figure 1, What is delta? “Delta shows the difference between the two formulations” What is the importance of delta?
Figure 2. Are these spheres form the same patient? How do the original, untreated spheres look like? In biology, the controls are extremely important, when there is an effect of solvent, which is the case here, the untreated control has to be shown.
Figure 3. add Cur A and CurB in the title of the graphs. There is no comment of panels A, B, D, G and I. they should be mentioned at least.
Lines 148-149 “In consequence, CurA treatment seems appropriate for other patient subgroups compared to CurB and vice versa.” Other patient subgroups?
Figures 1,3, 4, 8, 10 are a bit confusing. Usually inhibition is negative and stimulation is positive. I can understand the logic of current figure, but it is not apparent. If the Y axis is% viability then negative values are inhibition and positive are stimulation.
Why did the authors show individual samples? What is the diagnosis for these samples? Why did authors choose these samples and not others?
Material and Methods: How long did it take for spheres to form?
